# Emergence of novel methicillin resistant *Staphylococcus pseudintermedius* lineages revealed by whole genome sequencing of isolates from companion animals and humans in Scotland

Andrew R. Robb[1], Roisin Ure[1], Dominique L. Chaput[1], Geoffrey Foster[2]*

**1** Scottish Microbiology Reference Laboratories, Glasgow, United Kingdom, **2** SRUC Veterinary Services, Inverness, United Kingdom

* Geoffrey.Foster@sruc.ac.uk

## Abstract

*Staphylococcus pseudintermedius* is an opportunistic pathogen in dogs, and infection in humans is increasingly found, often linked to contact with dogs. We conducted a retrospective genotyping and antimicrobial susceptibility testing study of 406 *S. pseudintermedius* isolates cultured from animals (dogs, cats and an otter) and humans across Scotland, from 2007 to 2020. Seventy-five sequence types (STs) were identified, among the 130 isolates genotyped, with 59 seen only once. We observed the emergence of two methicillin resistant *Staphylococcus pseudintermedius* (MRSP) clones in Scotland: ST726, a novel locally-evolving clone, and ST551, first reported in 2015 in Poland, possibly linked to animal importation to Scotland from Central Europe. While ST71 was the most frequent *S. pseudintermedius* strain detected, other lineages that have been replacing ST71 in other countries, in addition to ST551, were detected. Multidrug resistance (MDR) was detected in 96.4% of MRSP and 8.4% of MSSP. A single MRSP isolate was resistant to mupirocin. Continuous surveillance for the emergence and dissemination of novel MDR MRSP in animals and humans and changes in antimicrobial susceptibility in *S. pseudintermedius* is warranted to minimise the threat to animal and human health.

## Introduction

*Staphylococcus pseudintermedius* is a major component of the normal skin flora in healthy domesticated dogs and is a frequent colonizer of the mucosae [1]. It can also act as an opportunistic pathogen, often implicated in skin, ear, urinary tract and post-surgical wound infections [2]. While not deemed to be part of the normal flora of a range of other companion animal species, transmission from dogs to cats, horses and birds has been shown [1].

Infection in humans caused by *S. pseudintermedius* are occasionally found; however, human infections can be underdiagnosed as *S. pseudintermedius* may be misidentified as *S.*

**Data Availability Statement:** 130 sequences available on Genbank. Accession numbers SAMN37310393 through to SAMN37310522.

**Funding:** This study was funded by the Veterinary Medicines Directorate. The funders had no role in study design, data collection and analysis, decision to publish, or preparation of the manuscript.

**Competing interests:** The authors have declared that no competing interests exist.

*aureus* or even coagulase negative staphylococci [3]. Human infections are considered to be zoonotic, linked directly or temporally to close contact with dogs [4,5].

Methicillin-susceptible *Staphylococcus pseudintermedius* (MSSP) have been shown to display high genetic diversity [6]. However, *S. pseudintermedius* resistant to methicillin (MRSP) have an epidemic clonal population structure, with four successful global lineages: ST71 in Europe, ST68 in North America, ST45 and ST112 in Asia [7]. Although the decline of predominance of the ST71 lineage was first reported in Northern European countries, in the last three years, several studies from other European countries have reported a similar trend, including Portugal, Italy and France [8–10].

MRSP has become a major issue in veterinary medicine, worsened by an increasing number of reports of multi and extensively antibiotic resistant strains that have drastically restricted therapeutic options in companion animals [11,12].

Determining the origins, evolutionary history and genetic basis of bacterial resistance from a genomic context is needed to explore patterns of distribution at a local and global level. The implementation of whole genome sequencing of veterinary pathogens has been instrumental in advancing the One Health concept, centred on the connection between animal, human and environmental health [13].

Data on the population structure and frequency of antimicrobial resistance of *S. pseudintermedius* in the UK is lacking. In this study, we assessed the genotypic diversity and antimicrobial susceptibility of *S. pseudintermedius* isolates from companion animals and humans, in Scotland, in a collection spanning a 14-year period.

## Materials and methods

### Bacterial isolates

Four hundred and six *S. pseudintermedius* isolates obtained from clinical samples were referred to the Scottish Microbiology Reference Laboratories, Glasgow (SMiRL) between 2007 and 2020. These isolates were from cats (8), dogs (366), humans (31) and an otter. Species identification was confirmed by an in-house real-time PCR (RT-PCR) using the Spseud primers and probe targeting *S. pseudintermedius* specific nuclease gene (Table 1). This assay was performed in 20μl reaction volumes containing primers and probes at the concentration

**Table 1. Sequences of oligonucleotides used for in-house real-time quadraplex PCR.**

| ID | Sequence 5' to 3' | Concentration (mM) | 5' modification | 3' modification |
|---|---|---|---|---|
| *mecA*_F | GGCAATATTAMCGCACCTCA | 0.3 | - | - |
| *mecA*_R | GTCTGCCASTTTCTCCTTGT | 0.3 | - | - |
| *mecA*_FAM | AGATCTTATGCAAACTTAATTGGCAAATCC | 0.2 | FAM | BHQ1 |
| SA-*nuc*_F | CATCCTAAAAAAGGTGTAGAGA | 0.3 | - | - |
| SA-*nuc*_R | TTCAATTTTMTTTGCATTTTCTACCA | 0.3 | - | - |
| SA-*nuc*_JOE | TTTTCGTAAATGCACTTGCTTCAGGACCA | 0.1 | JOE | BHQ1 |
| 23S_F | TACYCYGGGGATAACAGG | 0.8 | - | - |
| 23S_R | CCGAACTGTCTCACGACG | 0.8 | - | - |
| 23S_Cy5 | TTGGCACCTCGATGTCGG | 0.25 | Cy5 | BBQ650 |
| Spseud_F | TGAGCGCTTGAATCGATATT | 0.5 | - | - |
| Spseud_R | CACAGCCATATAGCCGCATTT | 0.5 | - | - |
| Spseud_TAMRA | AATGTCTGTTTCACTTTCA | 0.5 | TAMRA | BHQ2 |
| mupA_F | TAATGGGAAAATGTCTCGAGTAGA | 0.3 | TAMRA | BHQ2 |
| mupA_R | AATAAAATCAGCTGGAAAGTGTTG | 0.3 | | |
| mupA_TAMRA | CTCTATGCCGTTTGCTCAGCATCAT | 0.3 | | |

indicated in Table 1, 10μl QuantaBio PerfeCT multiplex qPCR SuperMix (QuantaBio, USA), 6μl nuclease free water and 4μl of template DNA from a purified 0.5 McFarland isolate inoculum (QiaSymphony, UK). Real-time PCRs were carried out on an ABI 7500 fast system (ThermoFisher, UK). Amplification consisted of initial warming at 50˚C for 5 minutes, *Taq* polymerase activation at 95˚C for 5 minutes followed by 30 cycles of 15s denaturation at 94˚C, 40s annealing/extension at 58˚C. The presence of *mecA* and *mupA* genes was detected by PCR and used to confirm methicillin and mupirocin resistance [14,15].

## DNA extraction, genome sequencing, assembly and annotation

A total of 130 isolates (85 MRSP and 45 MSSP) underwent whole genome sequencing. Genomic DNA was extracted using the semi-automated QIAsymphony AS platform following the manufacturer's protocol (Qiagen, Manchester, UK). DNA concentration was quantified using a Qubit fluorometer with the dsDNA High Sensitivity Kit (Invitrogen, Inchinnan, UK). DNA libraries were prepared using the Nextera XT DNA Library Preparation kit (Illumina, Cambridge, UK), according to the manufacturer's protocol, with 1 ng of genomic DNA per isolate. Sequencing was carried out on the Illumina MiSeq platform (Illumina, San Diego, CA, United States) with v3 chemistry, 2x300-bp paired-end kit, and 32 multiplexed isolates per run. Read quality was assessed using FastQC v 0.11.5 [16]. Reads passing the per base sequence quality and per sequence quality scores were included in further analysis. Raw reads were trimmed using Trimmomatic v 0.03 [17] and a sliding window cut-off of 15, de novo assembly was performed using Spades v 3.6.1 [18].

## MLST and resistance gene screening

In silico multi-locus sequence typing (MSLT) was performed using the sequence query tool of the *S. pseudintermedius* BIGbd website (https://pubmlst.org/spseudintermedius/).

Antimicrobial resistance genes were identified using two curated databases and associated search tools: the ResFinder 4.1 web server hosted by the Center for Genomic Epidemiology at https://cge.food.dtu.dk/services/ResFinder/ [19–21], and the CARD Resistance Gene Identifier, available at https://card.mcmaster.ca/analyze/rgi [22,23].

## Phylogenetic analysis

Phylogenetic relationships, based on the concatenated alignment of the high quality single nucleotide polymorphisms (SNPs), were inferred using CSI Phylogeny 1.4 from the Centre for Genomic Epidemiology, Denmark [24] with *S. pseudintermedius* E140 (NZ_ANOI00000000.1) as a reference genome and applying default settings [minimum depth at single-nucleotide polymorphism (SNP) positions: 10X; minimum relative depth at SNP positions: 10%; minimum distance between SNPs (prune): 10bp; minimum SNP quality: 30; minimum read mapping quality: 25; and minimum $Z$ score: 1.96]. The resultant tree was annotated using Interactive Tree of Life (iTOL) [25].

## Antimicrobial susceptibility testing

The antimicrobial resistance profile of all *S. pseudintermedius* isolates was determined, by disc diffusion on Mueller-Hinton agar (Oxoid, Wade Rd, Basingstoke, UK), following the Clinical Laboratory *Standards Institute* (CLSI) guidelines (VET01S, 2024) [26]. The following antimicrobials were tested; chloramphenicol (Ch) 30μg, clindamycin (Da) 2μg, enrofloxacin (En) 5μg, erythromycin (E) 15μg, gentamicin (Cn) 10μg, penicillin (P) 10μg, rifampicin (Rd) 5μg, trimethoprim 5μg (W), tetracycline (Te) 30μg (Oxoid, Wade Rd, Basingstoke, UK).

Interpretation of vancomycin susceptibility, using a 30μg disc, was based on previously described criteria for *S. aureus* by Rezaeifar et al [27]. Trimethoprim susceptibility was interpreted following the Clinical Laboratory Standards Institute (CLSI) guidelines (M100, 30th ed, 2020)[28]. Isolates that were resistant to three or more antimicrobial drug classes were defined as multi-drug resistant (MDR).

## Statistical analysis

Data manipulation, visualisation, and statistical analyses were carried out in R version 4.2.0 using tools from the tidyverse [29,30] and ComplexUpset [31] packages. Temporal changes in the proportion of specific sequence types (ST) were assessed using a generalised linear model with binomial distribution and year as the explanatory variable. Difference in the prevalence of phenotypic antibiotic resistance between MRSP and MSSP isolates was assessed using Fisher's exact test for count data (two-sided), with p-values corrected for multiple comparisons using the Bonferroni method to control the family-wise error rate. Values of $p < 0.01$ were considered statistically significant.

## Data availability

The sequence read dataset from this study were deposited in the European Nucleotide Archive under the project number PRJEB72695.

## Results

### Population structure of Scottish *S. pseudintermedius*

One hundred and thirty isolates, including all 85 MRSP from cat (4), dog (77), human (3) and a single otter and a representative selection of 45 MSSP from a cat (1), dog (19) and human (25) were selected for WGS (Fig 1A). Multi-locus sequence typing analysis identified seventy-five different sequence types (STs) with 59 (78.6%) seen only once. None of these STs were shared by both MRSP and MSSP populations. Only four STs, ST54, ST71, ST551 and ST1516, were detected in both animal and human *S. pseudintermedius* populations. Forty-nine STs (65.3%), 16 MRSP and 33 MSSP, were novel and sent to pubMLST for assignment (S1 Table).

Sequence type ST71 was the dominant MRSP clone in Scotland over the study period (28.2%), followed by ST68 (8.2%), ST726 (8.2%) and ST551 (4.7%) (Fig 1B). Together, these four STs accounted for 49.3% of all MRSP isolates. The MSSP had a more heterogeneous population structure with only three out of 42 STs, ST54, ST755 and ST1516, occurring more than once. Over the study period the prevalence of MRSP ST71 has appeared to decrease; it accounted for 50–66% of isolates in 2010 and 2014, but only 10–20% in 2018–2020 (though the decrease was not statistically significant, p = 0.433).

However, over this same time period, we observed the emergence of two new MRSP lineages, ST726 and ST551. ST726 was first identified in 2016 and has been since detected in seven dogs and ST551, the second lineage, identified in 2020, was detected in three dogs and one human.

### Core-genome SNP analysis

The genetic relatedness of Scottish MRSP and MSSP strains is visualised in a core-genome SNP (cgSNP) tree using *S. pseudintermedius* strain E140 as a reference genome (Fig 2). The SNP-based phylogenetic tree shows four main clades, corresponding to ST71, ST68, ST726 and ST551. Three of these clades, ST71, ST68 and ST551, are found on the same part of the tree and can be linked to a monophyletic branch. To support these observations, we used the pairwise SNP matrix which confirmed average SNP difference of 83 (range 3–133) for ST71,

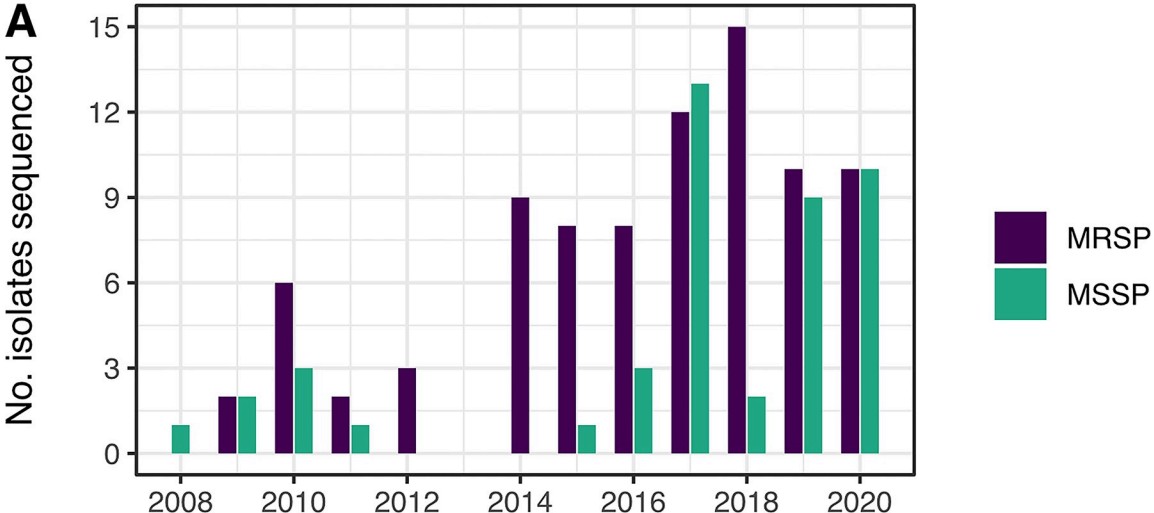

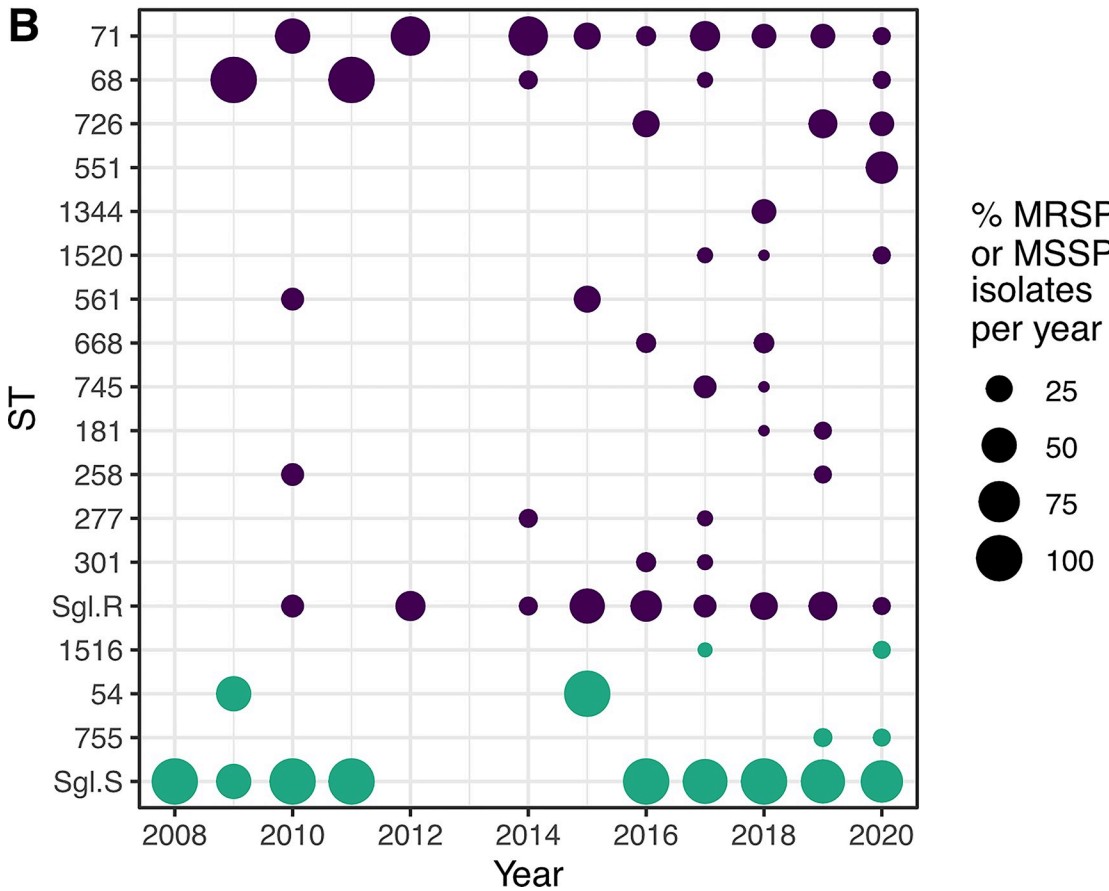

**Fig 1. Temporal variation of multi-locus sequence types over the study period.** (A) Number of MSSP and MRSP isolates that underwent whole genome sequencing, by year of collection, (B) Temporal variation in the relative abundance (%) of each ST, normalised against the total number of MRSP or MSSP isolates from that year. Sgl.R singleton MRSP; Sgl.S, singleton MSSP, i.e. STs observed only once across the entire data set.

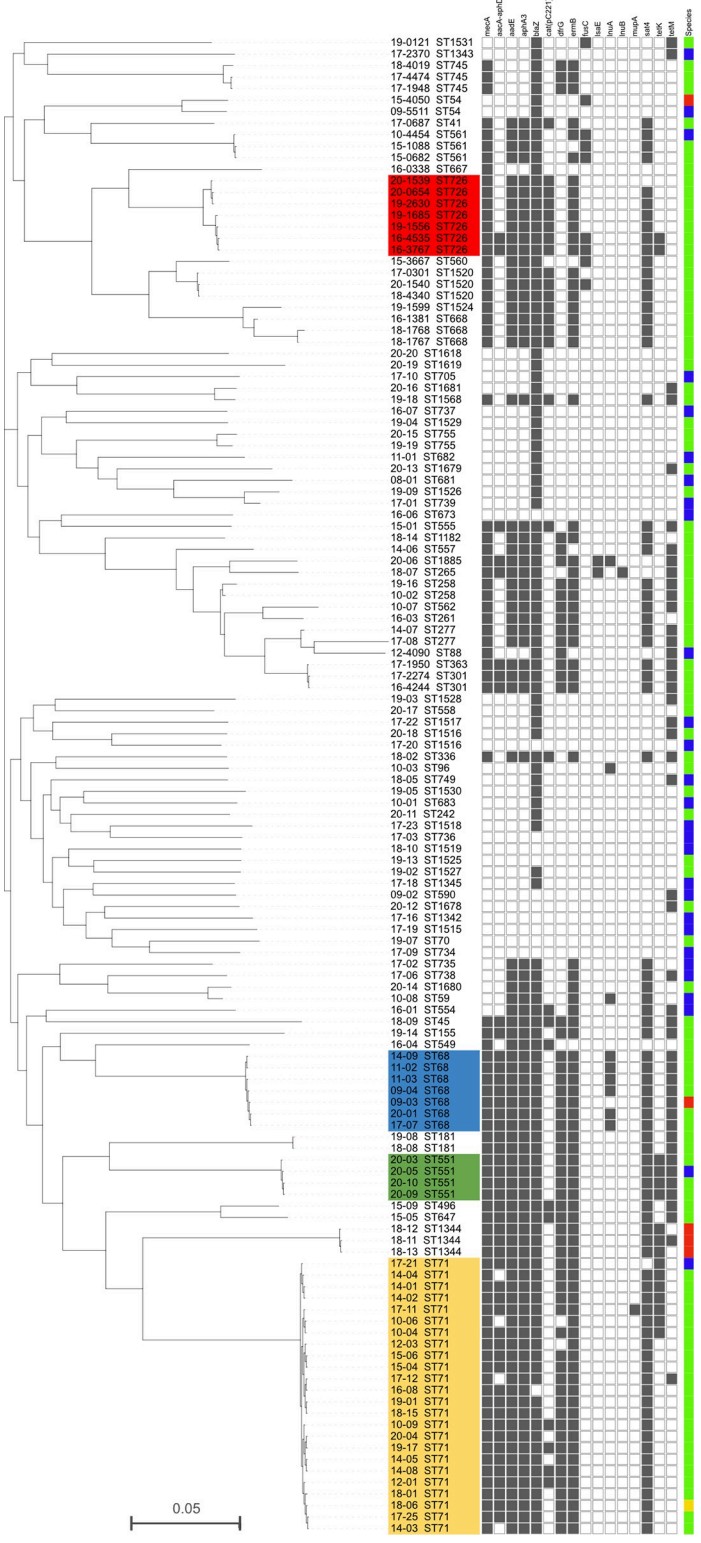

**Fig 2. Core-genome SNP tree of 130 *S. pseudintermedius* isolates that underwent whole genome sequencing.** The tip labels represent the strain name (beginning with the year of collection) followed by the ST. The colour strip indicates the host species: Green for dog, blue for human, red for cat and yellow for otter. The presence or absence of antimicrobial resistance genes are shown as either solid grey box for present and empty for absent. The length of each branch represents the number of SNP's. For clarity, the predominant clades are shown as coloured blocks; ST726 red, ST68 blue, ST551 green and ST71 yellow.

**Table 2. Frequency of antibiotic resistance among Scottish *Staphylococcus pseudintermedius* isolates.** Fisher's exact test for count data (two-sided) was used to compare resistance frequency in MSSP and MRSP isolates, with p-values corrected for multiple comparisons using the Bonferroni method.

| Antibiotic | All isolates (n = 406) | | MSSP (n = 321) | | MRSP (n = 85) | | *p* value |
|---|---|---|---|---|---|---|---|
| | Number | % | Number | % | Number | % | |
| Chloramphenicol | 33 | 8.1 | 5 | 1.6 | 28 | 32.9 | <0.001 |
| Clindamycin | 101 | 24.9 | 25 | 7.8 | 76 | 89.4 | <0.001 |
| Enrofloxacin | 57 | 14 | 3 | 0.9 | 54 | 63.5 | <0.001 |
| Erythromycin | 107 | 26.4 | 27 | 8.4 | 80 | 94.1 | <0.001 |
| Gentamicin | 83 | 20.4 | 3 | 0.9 | 80 | 94.1 | <0.001 |
| Mupirocin | 1 | 0.2 | 0 | 0 | 1 | 1.2 | 1.00 |
| Oxacillin | 85 | 20.9 | 0 | 0 | 85 | 100 | ND |
| Penicillin | 381 | 94 | 296 | 92.2 | 85 | 100 | 0.039 |
| Rifampicin | 0 | 0 | 0 | 0 | 0 | 0 | ND |
| Tetracycline | 59 | 14.5 | 12 | 3.7 | 47 | 55.3 | <0.001 |
| Trimethoprim | 142 | 35 | 74 | 23 | 68 | 80 | <0.001 |
| Vancomycin | 0 | 0 | 0 | 0 | 0 | 0 | ND |

58 for ST68 (range 5–88), 416 for ST726 (range 10–681) and for ST551 the average SNP difference was 63 (range 22–68). The observed phylogenetic diversity between MSSP genomes was high shown by the deep-branch structure corresponding with a distinct MLST.

## Antimicrobial susceptibility

A full set of susceptibility results for the 406 *S. pseudintermedius* isolates tested are shown in Table 2. Three-hundred and ninety (96.1%) were resistant to at least one of the antimicrobials tested. Resistance to rifampicin or vancomycin was not detected in any of the isolates. Resistance to penicillin was most common, detected in 381 (94%) isolates followed by trimethoprim 142 (35%), erythromycin 107 (26.4%) and clindamycin 101 (24.9%). A single canine isolate carried the *mupA* gene and was considered resistant to mupirocin. Eighty-five isolates carried the *mecA* gene and were considered oxacillin resistant.

Statistical analysis showed that MRSP had significantly higher rates of resistance to chloramphenicol, clindamycin, enrofloxacin, erythromycin, gentamicin, tetracycline and trimethoprim (p <0.001) than MSSP.

Forty-two different antimicrobial resistance phenotypes were detected of which 16 were seen only once. While 189 MSSP showed resistance exclusively to penicillin, the most frequent combination of antimicrobial resistance was penicillin-trimethoprim (n = 64) (S2 Table). The most frequently seen resistance phenotype among the MRSP was penicillin-oxacillin-gentamicin-enrofloxacin-erythromycin-clindamycin-tetracycline-trimethoprim (n = 21) (S3 Table). Multi-drug resistance (MDR), classified as resistance to ≥3 antimicrobial classes, was identified in 109 isolates (27%), 82 MRSP (96.4%) and 27 MSSP (8.4%), resulting in 32 MDR phenotypes, of which 14 were seen only once. Of the remaining 18 MDR phenotypes only four were detected in MSSP (Fig 3).

## Temporal variation in antimicrobial susceptibility

Antimicrobial resistance among the MRSP remained relatively stable across the study period. Among the 321 MSSP there was a decreasing temporal trend in resistance to trimethoprim from a high of 84% in 2015 to 8% in 2018 with no resistance detected in 2019 and 2020 and an increasing trend in tetracycline resistance from 1.3% in 2016 to 40% in 2020 (Fig 4). In non-ST71 MRSP, we observed a significantly lower rate of resistance to enrofloxacin, compared to

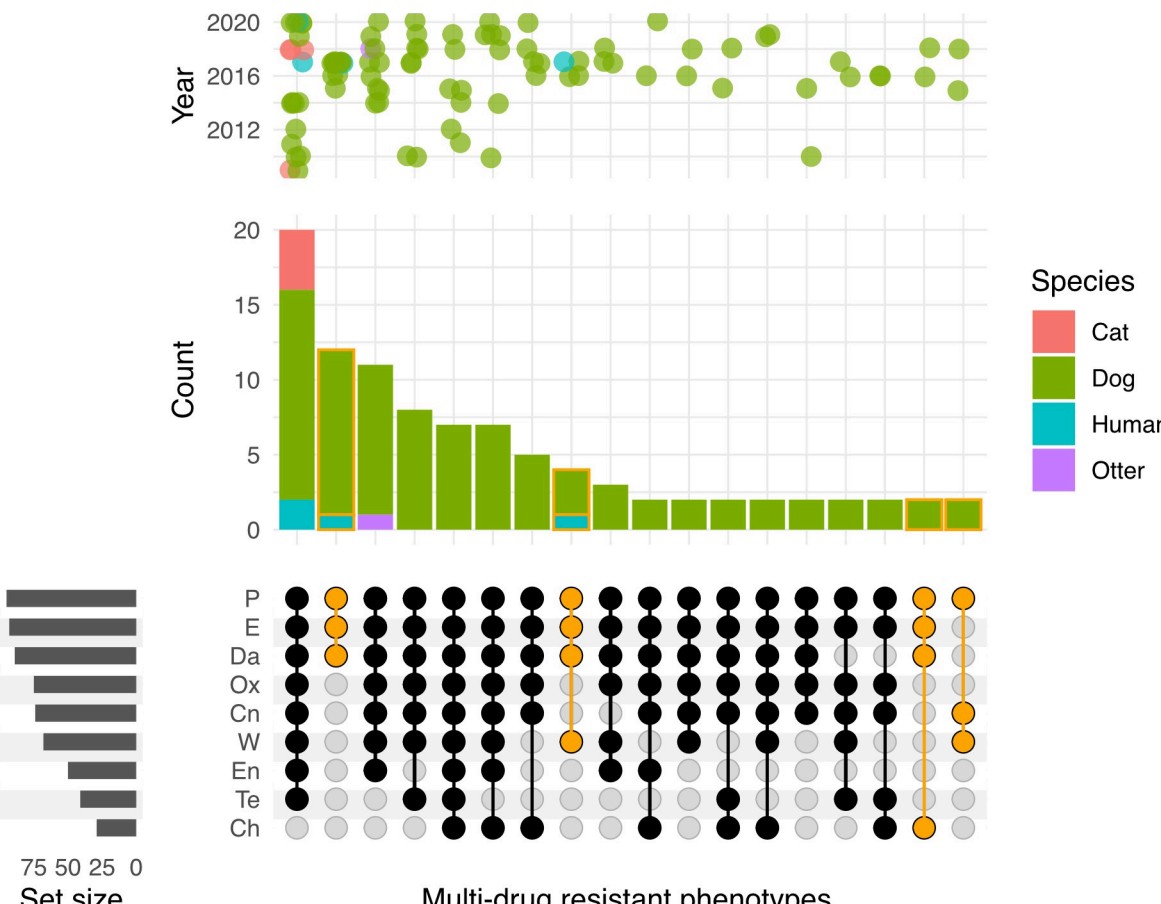

**Fig 3. Upset plot showing the intersections of resistance phenotypes among MDR *S. pseudintermedius* isolates.** Antimicrobials are indicated in the rows; P, penicillin; E, erythromycin; Da, clindamycin; Ox, oxacillin; Cn, gentamicin; W, trimethoprim; En, enrofloxacin; Te, tetracycline; Ch, chloramphenicol; while the intersections are shown in columns. Filled circles, MRSP black and MSSP yellow, indicate resistance and empty circles indicate susceptibility. The bar graph above indicates the number of isolates within the MDR phenotype and the bar chart to the left indicates the number of isolates resistant to the corresponding antimicrobial. The scatter plot shows the distribution of each MDR phenotype by year.

ST71 MRSP, over the study period (p <0.001). No other significant differences, in antimicrobial resistance, between these two groups was seen.

### Antimicrobial resistance genes

Antimicrobial resistance among the 130 isolates, selected for WGS, were attributed to the presence of the penicillin binding protein gene *mecA* (all β-lactams), the β-lactamase gene *blaZ* (penicillin), the tetracycline genes *tetM* and *tetK*, the aminoglycoside resistance genes *aacA-aphD*, *aadE* and *aphA3*, the macrolide, lincosamide and streptogramin B 23S rRNA methylase gene *ermB*, lincosamide nucleotidyltransferase *lnuA* and *lnuB*, the chloramphenicol acetyltransferase gene $cat_{pC221}$, trimethoprim-resistance dihydrofolate reductase gene *dfrG*, the fusidic acid resistance gene *fusC*, streptothricin acetyltransferase *sat4* and the mupirocin resistance gene *mupA*.

Six resistance genes, *aacA-aphD*, *dfrG*, *lsaE*, *lnuB*, *mupA* and *tetK*, were detected exclusively in MRSP, while among the MSSP isolates, none were carried exclusively. The distribution of AMR genotypes, among the 130 animal and human *S. pseudintermedius* strains, over the study period is shown in Fig 5.

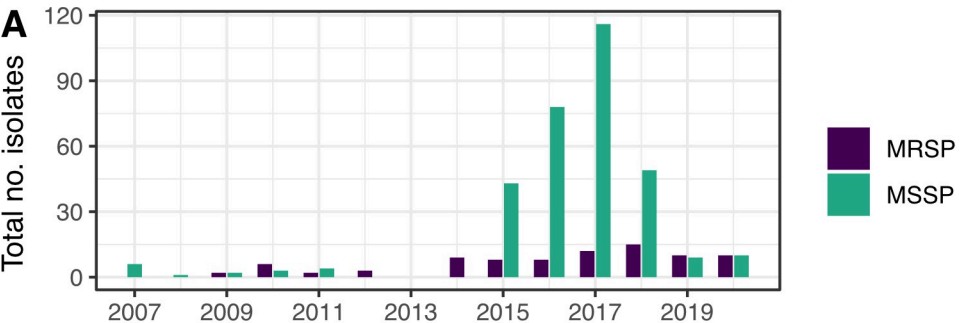

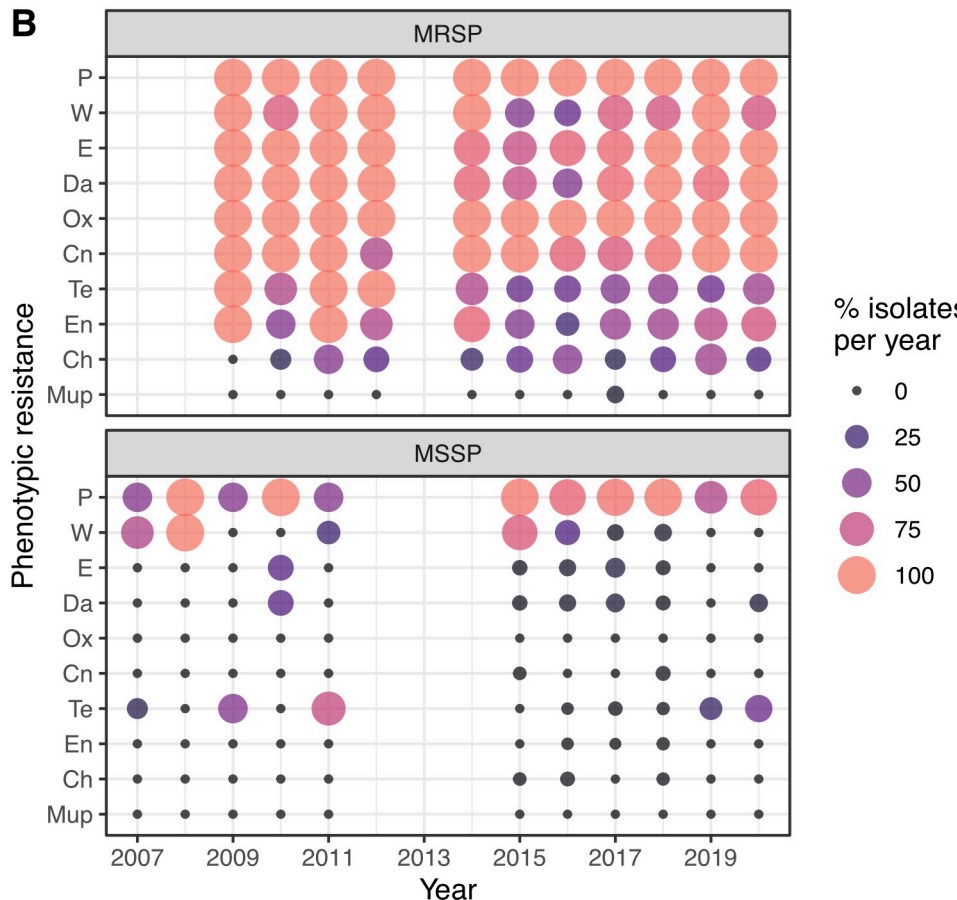

**Fig 4. Temporal differences in antimicrobial resistance rates of MRSP and MSSP isolates over the study period.**
(A) Number of MRSP and MSSP isolates that underwent susceptibility testing by year. (B) Temporal trends in phenotypic resistance, as a percentage of MRSP or MSSP isolates by year. Ch, chloramphenicol; En, Enrofloxacin; Cn, gentamicin; Da, clindamycin; E, erythromycin; Mup, mupirocin; Ox, oxacillin; P, penicillin; Te, tetracycline; Va, vancomycin and W, trimethoprim.

## Discussion

Whole genome sequencing, followed by MLST, identified 75 different STs among the 130 isolates examined in this study, with the majority occurring only once. This high overall genotypic diversity is notable and is similar to the situation observed in Denmark, Norway and the

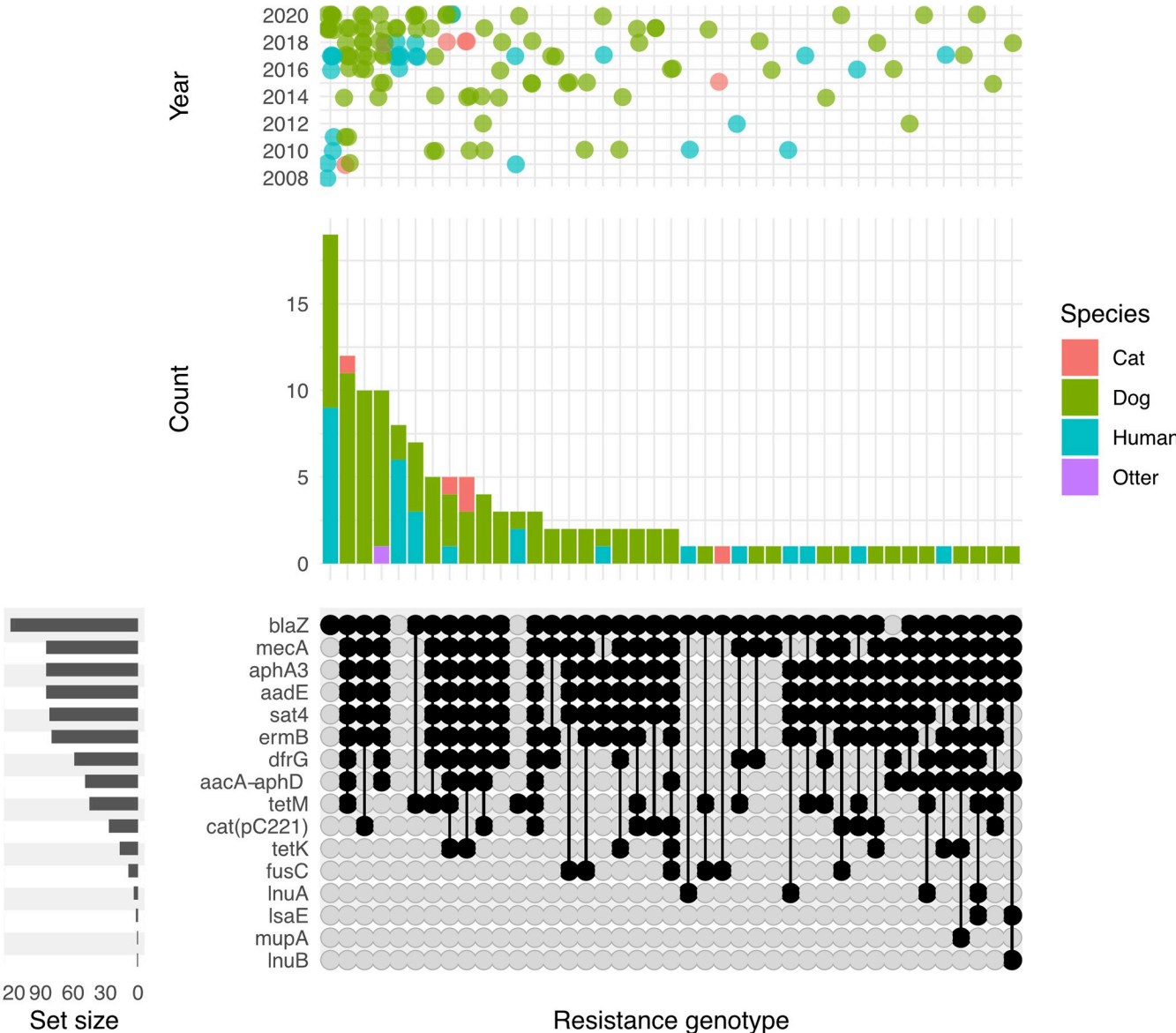

**Fig 5. Upset plot showing the intersections of resistance genotypes among 130 *S. pseudintermedius* isolates that underwent whole genome sequencing.** AMR genes are indicated in rows, while the intersections are shown in columns. Filled circles indicate gene presence and empty circles indicate absence. The bar chart above shows the number of strains within each resistance genotype and the bar chart to the left indicates the number of strains carrying the corresponding resistance gene. The scatter plot shows the distribution of each resistance genotype by year.

Netherlands for canine *S. pseudintermedius* [32]. However, MRSP has been spreading globally through the expansion of a small number of lineages with different genetic backgrounds. Population studies of *S. pseudintermedius* have shown four predominant clones; ST71 in Europe, ST68 in North America and ST45/ST112 in Asia, although, recent studies have indicated that the global population structure of MRSP is gradually changing and becoming more heterogeneous than previously recognised [33]. Additionally, data from studies undertaken in Italy, Portugal and France have shown an apparent decline in the prevalence of ST71, with the associated emergence of novel locally evolving clones [8–10]. In Norway, Denmark, Finland, the Netherlands, France and Italy, ST71 at present, remains the dominant clone, however, ST258

and related STs are becoming more prevalent across Europe [34]. Our results did not confirm these observations from other European countries, as only two isolates were recognised as ST258. A second MRSP clone, ST496, also appears to be challenging the dominance of ST71 in Australia, particularly around metropolitan Sydney and has also been detected in France [10]. In this study, we detected a single MDR ST496 isolated in 2015, but our results suggest, that this clone does not appear to be expanding in Scotland. The predominant clone, in our study, was ST71 accounting for 28.2% of strains, however, this prevalence does appear to be declining with a detection frequency of 50% in 2010, decreasing to 20% by 2020. Within our isolate collection we detected two lineages, ST551 and ST726 that have recently emerged in Scotland. MRSP ST551, initially discovered in Poland in 2015 and subsequently reported in Denmark, Sweden, Slovenia, Switzerland, South Africa, France, Portugal, Italy and the USA is also thought to be gradually replacing ST71 [7,10,32]. The four Scottish strains that belonged to ST551, three from dogs and one from a human, were all detected in the last year of this study. The other emerging lineage, ST726, was a novel ST, first identified in this study in 2016, with no other reports of this lineage reported in the literature. It is therefore likely, that this represents a locally evolving clone. A similar situation has been reported by Worthing and colleagues regarding the MDR MRSP clone ST496 in Australia [35]. The detection of two newly emerging clones in Scotland, may be a reflection of the shift to a more polyclonal MRSP population structure across Europe. It has been suggested that the observed change in the global population structure of MRSP can be partly attributed to differences in climate, dog population and density, the importation of dogs, as well as international travel may be possible contributing factors to this shift [36]. In recent years, there has been a large number of dogs imported to the UK from Eastern Europe and we suggest that this could be a possible route for the emergence of ST551 in the last year of our study.

Antibiotic-resistance remains one of the most important problems in the treatment and control of *S. pseudintermedius* related infection in human and veterinary medicine. In the present study only 16 (3.9%) isolates, of the 406 tested for susceptibility, were fully susceptible, a figure consistent with recent reports from Spain [37,38]. However, this frequency is much lower than data published from Chile, Portugal, France and Australia [6,8,39,40]. Overall, resistance to penicillin (94%) was the most common, followed by trimethoprim (35%), erythromycin (26.4%), clindamycin (24.9%) and gentamicin (20.4%). These figures are consistent with data on *S. pseudintermedius* resistance in studies from Germany, Finland, the Netherlands and France [6,41,42]. However, recent data have shown much higher rates of resistance, to these antimicrobials, in Italy, Brazil, Argentina, USA and South Africa [43–46]. Resistance to these agents in the UK is not surprising as they represent some of the most frequently sold antibiotics for the treatment of dogs and cats in Europe [47,48].This study did not find resistance to rifampicin or vancomycin in any of the isolates tested. While, rifampicin resistance has been reported, albeit at a very low level, in Australia, Denmark, Brazil and Italy, our results are consistent with those published by Ruscher et al, Viñes et al and Duim et al from Germany, Spain and the Netherlands respectively [34,38,49]. High rates of methicillin resistance and MDR in *S. pseudintermedius* and the approval of mupirocin for use in dogs, in South Korea and USA, has made mupirocin an attractive alternative for topical use in those countries for canine pyoderma. Mupirocin, however, is not licensed for use in dogs in the UK. In this study a single MDR *S. pseudintermedius* isolate, belonging to the dominant European lineage (ST71), isolated in 2017 from a dog, carried the *mupA* gene associated with high level mupirocin resistance (HLMR). This is the first report of HLMR in *S. pseudintermedius* in the UK. A similar low frequency of HLMR, in methicillin susceptible and resistant *S. pseudintermedius* from companion animals, has been reported in Croatia, Turkey, Poland, South Korea, Portugal and USA [8,50–52]. The plasmid encoded *mupA* gene is located in a mobile genetic

element, allowing horizontal movement of the gene between plasmids as well as vertical inheritance [53]. Of great concern is that plasmids carrying the *mupA* gene can also contain genes determining gentamicin, tetracycline and erythromycin resistance [54], therefore, exposure to any of these antimicrobials could co-select for HLMR strains. The fact that the *mupA* gene can be horizontally transferred, with the potential to co-select for multi-resistance, emphasises the importance of continuous monitoring for mupirocin resistance in *S. pseudintermedius* from companion animals and humans in the UK [53].

Resistance to chloramphenicol, clindamycin, enrofloxacin, erythromycin, gentamicin, tetracycline and trimethoprim was significantly higher among the MRSP isolates than MSSP isolates in our study. This high rate of resistance to antimicrobials in MRSP has also been shown in isolates from companion animals in Portugal, Finland and USA [8,42,55].

While our overall rates of resistance in MRSP and MSSP were similar to those reported from other European countries, it is the rate of MDR in both groups that is of concern. Susceptibility testing distinguished 42 different antimicrobial resistance phenotypes of which 76.2% were considered MDR. Multi-drug resistance was seen in both MRSP and MSSP populations, at 96.5% and 8.4% respectively. A similar situation has been reported by Maksimović et al in Bosnia-Herzegovina and Stefanetti et al in Italy [56,57].

Our data has revealed an overall steady rate of antimicrobial resistance in Scottish *S. pseudintermedius*, with the exception of trimethoprim which decreased from a high of 84% in 2015 to 8% in 2018 and an increasing trend in tetracycline resistance from 1.3% in 2016 to 40% in 2020. A similar increasing trend in tetracycline resistance, in the USA between 2010 and 2021, has also been observed by Phophi et al thought to be due to an increased use of doxycycline in the treatment of MRSP infections in dogs. However, they did not show an increase in the rate of resistance against folate inhibitors [58]. In contrast, a study by Lord et al has shown regional variation in antimicrobial resistance trends, in the USA, with a reduction in tetracycline and an increase in folate inhibitor resistance over a similar time period [55]. Our data show that non-ST71 MRSP present a significantly higher rate of susceptibility to enrofloxacin. Given the possible shift to a more polyclonal MRSP population structure in Scotland, the increased susceptibility to fluoroquinolones in non-ST71 increases the range of antimicrobials available for the treatment of *S. pseudintermedius* infection in animals. These results are consistent with data from France, where the introduction of an MRSP lineage other than ST71 resulted in an increase in susceptibility to a number of antimicrobials, including fluoroquinolones [10]. Conversely, in a report from the USA, covering a similar time period as this study, while they show a lower rate of fluoroquinolone resistance in MRSP this rate remained stable [55].

Possible reasons behind the change in trimethoprim and tetracycline resistance are unclear, but could include a change in the use of antimicrobials or changes of compound in companion animals in Scotland. Unfortunately, information on antimicrobial usage in companion animals in Scotland are composed from only a few veterinary practices limiting the usefulness of their application in our study [59]. UK figures, however, are available that show that over the period 2014–2020, usage showed a downward trend for most reported antibiotics in dogs [47].

Among the 130 isolates selected for WGS, 16 antimicrobial resistance genes were identified. None were exclusively carried by MSSP, however, the genes encoding gentamicin (*aacA-aphD*), trimethoprim (*dfrG*), lincosamide (*lnuB*, *isaE*) and tetracycline (*tetK*) resistance were exclusively carried by MRSP which is consistent with data published by Ruiz-Ripa et al [37]. A number of these resistance genes are carried on mobile genetic elements posing a threat in the treatment of human and animal *S. pseudintermedius* infections [60].

Overall, there was a high level of concordance between phenotypic and genotypic antimicrobial resistance which was consistent with data published for *S. pseudintermedius* by Tyson et al and Viñes et al [38,61]. Antimicrobial susceptibility testing by disc diffusion therefore

appears to be a reliable method for the determination of resistance in *S. pseudintermedius* isolates.

In conclusion, this is the first report of the population structure and the frequency of antimicrobial resistance of *S. pseudintermedius* from animals and humans in the UK. Our data shows an apparent decline in the dominant European MRSP clone, ST71, with the concomitant emergence of two MRSP lineages; novel, locally evolving MRSP clone ST726 and an ST551 MRSP lineage that may have occurred by animal importation from central Europe. While the decline in the prevalence of ST71 has been recognised in a number of European countries, it remains unclear which other MRSP clone or clones will be its successor.

We have also shown temporal variation in the frequency of resistance to antimicrobials used in human and veterinary medicine. It is therefore, important that we monitor closely the emergence and dissemination of novel MDR *S. pseudintermedius* strains in the UK and the changes in antimicrobial susceptibility to minimise the threat to animal and human health.

## Supporting information

**S1 Table. Novel STs submitted to pubMLST for assignment.**
(DOC)

**S2 Table. Full set of resistance phenotypes for MSSP (n = 321).**
(DOC)

**S3 Table. Full set of resistance phenotypes for MRSP (n = 85).**
(DOC)

## Acknowledgments

We acknowledge the laboratory staff at SRUC Veterinary Services and Scottish Health Boards for submitting *S. pseudintermedius* isolates to the Scottish Microbiology Reference laboratories, Glasgow. We also thank Professor M. Holden for his advice on the interpretation of the cgSNP phylogenetic tree and the reviewers for their helpful comments, which improved this manuscript.

## Author Contributions

**Conceptualization:** Andrew R. Robb, Geoffrey Foster.

**Data curation:** Andrew R. Robb, Dominique L. Chaput, Geoffrey Foster.

**Formal analysis:** Andrew R. Robb, Dominique L. Chaput, Geoffrey Foster.

**Funding acquisition:** Andrew R. Robb, Geoffrey Foster.

**Investigation:** Andrew R. Robb, Roisin Ure, Geoffrey Foster.

**Methodology:** Andrew R. Robb, Roisin Ure, Geoffrey Foster.

**Project administration:** Andrew R. Robb, Geoffrey Foster.

**Resources:** Geoffrey Foster.

**Software:** Dominique L. Chaput.

**Supervision:** Andrew R. Robb, Geoffrey Foster.

**Validation:** Andrew R. Robb, Roisin Ure.

**Writing – original draft:** Andrew R. Robb, Roisin Ure, Geoffrey Foster.

**Writing – review & editing:** Andrew R. Robb, Roisin Ure, Dominique L. Chaput, Geoffrey Foster.

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
