## [Decision Letter · Decision Letter 0]

20 Dec 2023

PONE-D-23-29835Emergence of a novel methicillin resistant Staphylococcus pseudintermedius strain and likely importation of another from Europe revealed by whole genome sequencing of Staphylococcus pseudintermedius from companion animals and humans in ScotlandPLOS ONE

Dear Dr. Foster,

Thank you for submitting your manuscript to PLOS ONE. After careful consideration, we feel that it has merit but does not fully meet PLOS ONE’s publication criteria as it currently stands. Therefore, we invite you to submit a revised version of the manuscript that addresses the points raised during the review process.

The manuscript has been assessed by one reviewer and by myself.  Comments are available below. The reviewer has raised a number of concerns about the methodology and the data, he/she recommends revisions to provide a fuller outline of the methodology and main results.

Please carefully revise the manuscript to address all the points raised by the reviewer.

We look forward to receiving your revised manuscript.

Kind regards,

Jose Melo-Cristino, M.D., Ph.D.

Academic Editor

PLOS ONE

3. Please upload a new copy of Figure 1 as the detail is not clear. Please follow the link for more information: https://blogs.plos.org/plos/2019/06/looking-good-tips-for-creating-your-plos-figures-graphics/" https://blogs.plos.org/plos/2019/06/looking-good-tips-for-creating-your-plos-figures-graphics/

Reviewers' comments:

Reviewer's Responses to Questions

**Comments to the Author**

1. Is the manuscript technically sound, and do the data support the conclusions?

Reviewer #1: Partly

2. Has the statistical analysis been performed appropriately and rigorously? 

Reviewer #1: Yes

3. Have the authors made all data underlying the findings in their manuscript fully available?

Reviewer #1: Yes

4. Is the manuscript presented in an intelligible fashion and written in standard English?

Reviewer #1: Yes

5. Review Comments to the Author

Reviewer #1: The manuscript describes the characterization of a wide collection of Staphylococcus pseudintermedius isolates collected During 14 years in Scotland. Antimicrobial susceptibility profiles were determined for all isolates. A subset of these isolates, including all MRSP detected, were subjected to whole genome sequencing. Staphylococcus pseudintermedius is a relevant opportunistic pathogen in veterinary medicine, for which there has been alarming reports concerning increasing AMR. Therefore, updated data on S. pseudintermedius population structure and AMR is urgently needed.

The manuscript is well written and structured. However, some aspects need clarification.

A major concern regards the criteria used for the antimicrobial susceptibility profiling of the isolates (Lines 155-173). It is stated in line 110 (materials and methods) that antimicrobial susceptibility was performed according to EUCAST recommendations. However, EUCAST does not establish zone diameter breakpoints for S. pseudintermedius and several of the antimicrobials tested in this study, such as enrofloxacin (not included in EUCAST recommendations), teicoplanin and vancomycin (glycopeptides only have MIC breakpoints defined). Also, penicillin zone diameter breakpoints are only established for S. aureus and S. lugdunensis. In addition, for mupirocin, only ECOFF values are given for S. aureus when testing a 200 ug mupirocin disc (EUCAST, 2022, v12.0, page 104). Therefore, clarification is needed for readers to understand the categorization of isolates as S or R. Also, in light of these observations, it is suggested a revision of the statement in lines 314-315.

The manuscript describes in detail the population structure of the Scottish S. pseudintermedius, yet its contextualization with AMR phenotypes and genotypes is limited. For instance, similarly to what is occurring in other European countries, a change in the population structure was observed but no discussion on its potential (or absent) impact on AMR is made. For example, are the new lineages associated with different antibiotic resistance phenotypes/genotypes, as already proposed in other studies?

It is suggested that the title of the manuscript be shortened, such as "Emergence of novel methicillin resistant Staphylococcus pseudintermedius lineages revealed by whole genome sequencing of Staphylococcus pseudintermedius from companion animals and humans in Scotland".

Other minor concerns/suggestions are detailed below:

- Line 32: The study by Ruscher et al. (ref #1) focus on the identification and characterization of coagulase positive staphylococci, including S. pseudintermedius, from only infection sites in dogs, cats and Equidae; thus, other study/review could be cited to support the statement of the status of S. pseudintermedius as a colonizer of dogs, such as ref #3.

- Lines 36 and 39: Please consider citing the review by Blondeau et al. Zoonotic Staphylococcus pseudintermedius: an underestimated human pathogen? Future Microbiol. 2023 Apr;18:311-315. doi: 10.2217/fmb-2023-0069.

- Lines 40 – 42/210-217: This statement is true for MRSP, but not for MSSP for which a wide genetic diversity has been reported in several studies (review ref #3; further corroborated in more recent studies such as the ones by Haenni et al. J Glob Antimicrob Resist. 2020;21:57-59. doi: 10.1016/j.jgar.2020.02.016. and Morais et al. Front Microbiol. 2023;14:1167834. doi: 10.3389/fmicb.2023.1167834.). In addition, I believe ref #8 was exchanged with ref #7.

- Lines 42 -45/210-217: It is suggested an update on this statement. Although the decline of predominance of ST71 lineage was first reported in Northern European countries (e.g., ref# 22; ref# 17; ref# 20); in the last three years, several other studies from other European countries have reported a similar trend, including Italy (Nocera et al. Pol J Vet Sci. 2020;23:465-468. doi: 10.24425/pjvs.2020.134693), France (ref# 19) and Portugal (Morais et al. Front Microbiol. 2023;14:1167834. doi: 10.3389/fmicb.2023.1167834).

- Line 64: It is not clear from the text and Table 1, which target(s) were used for species identification. Please disclose the target(s) used in the main text.

- Line 72: The primers for mupA detection appear to be missing from Table 1.

- Line 89: Please detail the type of assembly. Did the authors followed a de novo strategy or performed the assembly against a reference strain ?

- Lines 109-114: EUCAST recommendations do not have breakpoints

- Lines 116-117: Please disclose the software used for statistical analysis.

- Line 121 - Please disclose the source of the 85 MRSP isolates.

- Line 122 - Was there any criteria for the selection of the MSSP isolates for WGS?

- Line 128 - Are there any STs shared by both MRSP and MSP populations?

- Lines 131-133: What was the trend of ST71 detection throughout the time?

- Line 132: “… and has been [since] detected in seven dogs.”

- Line 132. Please correct to “The second lineage, …”

- Lines 137- 138: This statement needs clarification. As authors state, analysis of Figure 1 shows that the Scottish S. pseudintermedius isolates can be divided in four main clades. However, each clade is not represented by each ST mentioned. For example, ST71, ST68 and ST551 appear to be part of the same clade, independently of the homogeneity degree of the strains that constitute each ST.

- Line 146, legend of Figure 1: As mentioned previously, the otter isolate was not subjected to the WGS. Therefore. there are no isolates labeled as yellow.

Tables 3 and 4: It is suggested a different system for organization of antimicrobial resistance phenotypes, either by total number of isolates or by increasing resistance to antimicrobials. From previous results it is deduced that the otter isolate corresponds to a MSSP. However, this isolate (or animals species) does not appear in table 3 or in table 4.

Line 172: The frequency of MDR isolates should be given also within MRSP and MSSP isolates.

Lines 198 – 206: the description of results regarding the presence of antimicrobial resistance genes is poor and should be more detailed. For example, there is no analysis on the distribution of AMR genes and clonal lineages or animal species.

Figure 2: The names qacC, qacD (more rarely) and smr are given in literature to the same gene (https://tcdb.org/search/result.php?tc=2.A.7.1.1), which encodes an efflux pump (Smr/QacC/QacD) from the SMR family of transporters, associated with reduced susceptibility to biocides. This gene is located in plasmids and its frequency in S. pseudintermedius is not as high (e.g., Hritcu et al. Front Vet Sci. 2020;7:414. doi: 10.3389/fvets.2020.00414) as the reported in this study. It is suggested that authors verify the frequency of these genes in the genomes analyzed, since it could be related to gene annotations “errors”. For example, a gene could have been annotated as qacC/qacD because it belongs to the SMR family (qacC-like) and not for its high identity with the smr/qacC/qacD gene; in this case, the result should be interpreted with caution.

- Line 229: “we detected two lineages, ST551 and ST726, …”

- Line 237: “… no other reports of this lineage…”

- Line 240/241: “… may be a reflection of the shift…”

- Line 230: ST551 strains have additionally been reported in France (ref #19), Italy (Vitali et al. J Glob Antimicrob Resist. 2021;25:107-109. doi: 10.1016/j.jgar.2021.02.025) and Portugal (Morais et al. Front Microbiol. 2023;14:1167834. doi: 10.3389/fmicb.2023.1167834).

- Lines 253-255/280-292: Relevant studies have also been published in the last two years. It is suggested an update on the statements and references.

-Lines 293-295: The graphical presentation of some of trends of frequency of AMR (including MRSP/MDR) throughout the 14 years of study would be of interest.

- In the references list, there are some italics missing (for example, in lines 346 and 376)

6. PLOS authors have the option to publish the peer review history of their article (what does this mean?). If published, this will include your full peer review and any attached files.

Reviewer #1: No

---

## [Author Response · Author response to Decision Letter 0]

5 Apr 2024

A comprehensive response has been provided in the rebuttal

---

## [Decision Letter · Decision Letter 1]

16 May 2024

PONE-D-23-29835R1Emergence of novel methicillin resistant Staphylococcus pseudintermedius lineages revealed by whole genome sequencing of isolates from companion animals and humans in ScotlandPLOS ONE

Dear Dr. Foster,

Thank you for submitting your manuscript to PLOS ONE. After careful consideration, we feel that it has merit but does not fully meet PLOS ONE’s publication criteria as it currently stands. Therefore, we invite you to submit a revised version of the manuscript that addresses the points raised during the review process. Your manuscript has been returned to the original reviewer  and a minor revision is still suggested.

We look forward to receiving your revised manuscript.

Kind regards,

Yung-Fu Chang

Academic Editor

PLOS ONE

Journal Requirements:

Reviewers' comments:

Reviewer's Responses to Questions

**Comments to the Author**

1. If the authors have adequately addressed your comments raised in a previous round of review and you feel that this manuscript is now acceptable for publication, you may indicate that here to bypass the “Comments to the Author” section, enter your conflict of interest statement in the “Confidential to Editor” section, and submit your "Accept" recommendation.

Reviewer #1: (No Response)

2. Is the manuscript technically sound, and do the data support the conclusions?

Reviewer #1: Yes

3. Has the statistical analysis been performed appropriately and rigorously? 

Reviewer #1: Yes

4. Have the authors made all data underlying the findings in their manuscript fully available?

Reviewer #1: Yes

5. Is the manuscript presented in an intelligible fashion and written in standard English?

Reviewer #1: Yes

6. Review Comments to the Author

Reviewer #1: The authors have answered all my queries and revised the manuscript accordingly. However, there are still some minor details that I think need clarification or revision. They are listed below:

Materials and methods:

Subsection “Antimicrobial susceptibility testing”:

- a few more details should be given, such as suppliers of culture media and antibiotic discs;

- the CLSI guidelines VET01S (2024) does not establish breakpoints for trimethoprim (only for trimethoprim-sulphamethoxazole). Please clarify which guidelines were used for categorization.

- Please disclose the vancomycin disc content (it is an information also absent in the cited article).

Results:

Lines 147 and 155: Please refer to Figure 1-A and Figure 1-B, respectively.

Line 247: “… the tetracycline resistance genes tet(M) and tet(K)…” It is also suggested a revision of tet gene nomenclature throughout the manuscript.

Line 252: missing italic in “sat4”

Line 296: “… one from a human …”

Line 297: identified for the first time in 2017 or 2016 ?

Line 329: I suggest that authors nuance the statement “… and expressed high level mupirocin resistance.”, since authors did not determine mupirocin MICs and are only inferring high level resistance through detection of mupA. I suggest “… carried the mupA gene, associated with high level mupirocin resistance (HLMR).”

Line 333: I suggest removing the text “[which corresponds to HLMR]”

Line 335: I suggest revision of the sentence to “ … horizontal transfer of mupA between different staphylococcal species.”

Figure 1: Although visually appealing, I am not sure I completely understand Figure 1-B. For instance, looking at year 2010, four circles of MRSP strains are shown corresponding to % higher than 100 (50 + 25 + 25+ 25). I suggest changing the legend to “Percent of MRSP and MSSP isolates per year”; also there is a circle size used in the Figure that is not included in the legend.

7. PLOS authors have the option to publish the peer review history of their article (what does this mean?). If published, this will include your full peer review and any attached files.

Reviewer #1: No

---

## [Editor Report · Decision Letter 2]

27 May 2024

Emergence of novel methicillin resistant Staphylococcus pseudintermedius lineages revealed by whole genome sequencing of isolates from companion animals and humans in Scotland

PONE-D-23-29835R2

Dear Dr. Foster,

We’re pleased to inform you that your manuscript has been judged scientifically suitable for publication and will be formally accepted for publication once it meets all outstanding technical requirements.

Kind regards,

Yung-Fu Chang

Academic Editor

PLOS ONE
---

## [Editor Report · Acceptance letter]

25 Jun 2024

PONE-D-23-29835R2 

PLOS ONE

Dear Dr. Foster, 

I'm pleased to inform you that your manuscript has been deemed suitable for publication in PLOS ONE. Congratulations! Your manuscript is now being handed over to our production team.

Kind regards, 

on behalf of

Dr. Yung-Fu Chang 

Academic Editor

PLOS ONE